# Comparison of an Artificial Neural Network and a Response Surface Model during the Extraction of Selenium-Containing Protein from Selenium-Enriched *Brassica napus* L.

**DOI:** 10.3390/foods11233823

**Published:** 2022-11-27

**Authors:** Tao Yang, Hongkun Lai, Zimo Cao, Yinyin Niu, Jiqian Xiang, Chi Zhang, Longchen Shang

**Affiliations:** 1College of Biological and Food Engineering, Hubei Minzu University, Enshi 445000, China; 2Enshi Tujia & Miao Autonomous Prefecture Academy of Agricultural Sciences, Enshi 445000, China

**Keywords:** selenium-containing protein, BP artificial neural network, genetic algorithm, bio-activity, selenium-enriched rape

## Abstract

In this study, the extraction conditions for selenium-enriched rape protein (SEP) were optimized by applying a response surface methodology (RSM) and artificial neural network (ANN) model, and then, the optimal conditions were obtained using a genetic algorithm (GA). Then, the antioxidant power of the SEP was examined by using the DPPH, ABTS, and CCK-8 (Cell Counting Kit-8), and its anticancer activities were explored by conducting a cell migration test. The results showed that compared with the RSM model, the ANN model was more accurate with a higher determination coefficient and fewer errors when it was applied to optimize the extraction method. The data obtained for SEP using a GA were as follows: the extraction temperature was 59.4 °C, the extraction time was 3.0 h, the alkaline concentration was 0.24 mol/L, the liquid-to-material ratio was 65.2 mL/g, and the predicted content of protein was 58.04 mg/g. The protein was extracted under the conditions obtained by the GA; the real content of protein was 57.69 mg/g, and the protein yield was 61.71%. Finally, as the concentration of the selenium-containing protein increased, it showed increased ability in scavenging free radicals and was influential in inhibiting the proliferation and migration of HepG2 cells.

## 1. Introduction

Selenium (Se) is an essential trace element for animals and humans and has attracted academic and commercial attention due to its bio-active functions, including antioxidant bio-activity, anticancer bio-activity, cardiovascular conditioning, etc. [1]. It has been estimated that at least 70% of land area in China is deficient in selenium [2]. Reportedly, selenium deficiency causes a variety of diseases; Kashin–Beck and Kashin diseases are typical selenium deficiencies [3]. Additionally, Yao et al. [4] found that the substitution of sulfur for selenium in the GPX4 active center of the mouse embryonic cells resulted in major iron death, a form of programmed cell death that is driven by the iron-dependent accumulation of lipid peroxidation, in neuronal cells. Their research result was a strong piece of evidence showing that selenium is essential for mammals. Moreover, as consumers’ increase in health awareness, the importance of dietary selenium is gradually being recognized. It is well known that Se supplements can usually be divided into organic and inorganic forms, while selenium-containing proteins (SCP) belong to the former category. Selenium is absorbed by plants and participates in protein synthesis instead of sulfur [5]. Numerous kinds of research have shown that inorganic Se supplements are of a wider safe-dose range and have higher bio-availability compared to the inorganic form [5,6]. Mahammad et al. [7] found that the yolk color and antioxidant activity of eggs laid by hens fed SCP was significantly improved, and the shelf life of eggs was also extended. Furthermore, Song et al. [8] found that glucose deprivation-induced apoptosis in neuronal cells could be inhibited when selenium-enriched spirulina proteins were provided. Their research demonstrated a potential application of selenium-containing proteins in the chemoprevention and treatment of ischemic brain injury in humans. Moreover, a specific SCP was found to be highly related to the formation of abnormal crypt foci (ACFs) in the colon cancer tumor cells of mice induced by azomethine [9]. Altogether, the growing number of studies have revealed the development and application prospects of SCP in medicine and functional food.

The importance of efficiently extracting SCPs for studying their functional properties is self-evident. Currently, solvent extraction, enzymatic methods, etc., are the commonly used research methods to extract the protein, and these methods are usually assisted by ultrasound or freeze–thaw treatment. For example, Gao et al. [10] extracted selenium-containing protein from selenium-enriched peanuts using the solvent extraction method with a protein yield of 87%. However, there are still some problems with the enzymatic and ultrasound-assisted methods. For the former method, its extraction yield has usually been relatively low, while the protein could easily be denatured when the latter method was applied, which is a significant challenge in SCP industrial production [2,11].

Orthogonal experimental design (OED), response surface methodology (RSM), and artificial neural network (ANN) are the primary methods of optimization used in protein extraction, as well as SCP extraction. Both the application of OED and RSM have been relatively more common than the ANN, and the predictive accuracy of OED have been quite poor, while the RSM has shown a higher prediction accuracy and has been capable of more workload [12]. However, the RSM is mainly optimized by a multivariate quadratic model, and its prediction accuracy is also influenced by a non-fitting factor [13]. ANN was one of the artificial intelligence techniques that explored complicated systems’ relationships by fitting the existing data and back-propagating them [14]. In recent years, ANN has also been widely used to solve non-linear and nonstationary conditions because of its high accuracy of fitted and predicted values [15]. Wang et al. [16] showed that an ANN model optimized the index of millet pasting characteristics, and the lowest mean square error value of 0.0175 indicated that the model was favorably generalized. Aklilu et al. [17] found that trained ANN models were more accurate than RSM in their predictive ability.

In this study, the extraction process of selenium-enriched rape protein (SEP) was optimized based on single-factor experiments. The differences in error values, prediction data, and optimal solutions between the RSM model and ANN model were compared, and the SEP activity was finally explored. The genetic algorithm (GA) was used to perform a global optimization of this function since the prediction model obtained by ANN was a nonlinear function in input and output. We hope that the result of this study will provide a new research perspective on SCP extraction.

## 2. Materials and Methods

### 2.1. Materials and Chemicals

Selenium-enriched rape (*Brassica napus* L.) was provided by Enshi Tujia and the Miao Autonomous Prefecture Academy of Agricultural Sciences (Enshi, Hubei, China).

The following chemicals and reagents were used in this study: bovine serum albumin, potassium borohydride, trichloroacetic acid, Coomassie brilliant blue G250 and were purchased from Sinopharm Chemical Reagent Co., Ltd. (Shanghai, China); the 3.5 kDa dialysis bag was purchased from Beijing Ruida Henghui Technology Development Co., Ltd. (Beijing, China); the selenium single element solution standard GBW(E)-080215 was purchased from the China Academy of Metrology (Beijing, China); 2, 2-diphenyl-1-picrylhydrazyl (DPPH) and 2, 2′-azino-bis(3-ethylbenzothiazoline-6-sulfonic acid) (ABTS) were purchased from Sigma-Aldrich Co. (St. Louis, MO, USA); the HepG2 cells were purchased from Wuhan Pronosai Life Sciences Co. (Wuhan, China); fetal bovine serum, penicillin + streptomycin double antibodies were purchased from Lanjiike Technology Co., Ltd. (Biosharp, Anhui, China); fluorouracil (5-Fu) was purchased from MedChem Express Co., Ltd. (Princeton, NJ, USA); 0.25% trypsin and medium (DMEM high glucose) were purchased from Invitrogen Co. (Gibco, New York, NY, USA); Enhanced CCK-8 (Cell Counting Kit-8) was purchased from Beyotime Biotechnology Co., Ltd. (Shanghai, China); other chemicals were also purchased from Sinopharm Chemical Reagent Co., Ltd. (Shanghai, China).

### 2.2. Protein Extraction

#### 2.2.1. Preparation of Selenium-Enriched Rape Powder

The selenium-enriched rape (SER) with root removed was cleaned with water and placed in a blast drying oven (GXZ-9140 Shanghai Boxun Industrial Co., Ltd., Shanghai, China) at 50 °C for 48 h. The SER was ground by a crusher (BJ-1000A, Hangzhou Baijie Technology Co., Ltd., Hangzhou, China). Then, the powder of SER was filtered through an 80-mesh sieve and stored in a desiccator for spare.

#### 2.2.2. Selection of Extraction Solvent and Five Single-Factor Experiments Design

The protein content in the solution was determined by referring to the method of Snyder et al. [18], and the standard curve of bovine serum protein was plotted as Y = 0.0058X − 0.0047, where Y represents the determined absorbance, X represents the protein content in the solution (unit: mg/g). The correlation coefficient of this curve was 0.9973.
(1)$$\mathrm{Protein}\;\mathrm{content}\;\left(\mathrm{mg}/g\right)=\frac{a}{m}$$
where a represents the protein content in the supernatant, m represents the mass of SER powder.

The extraction solvents were selected by using the method of Duangjarus et al. [19] with slight modifications. An amount of 1g SEP powder was filled into each of five centrifuge tubes before 10 mL of ultra-pure water, NaCl solution (1 mol/L), ethanol solution (75% by volume), alkaline solution (0.1 mol/L NaOH solution), and acid solution (0.1 mol/L HCl solution), respectively, were added into those tubes. Then, the tubes were stirred by magnetic stirrers (B11-2, Shanghai Lingyi Biotechnology Co., Ltd., Shanghai, China) at 50 °C for 1 h before centrifuging at 6793 g and 4 °C for 15 min. The protein content in the system was determined by a UV-vis spectrophotometer (UV9000S, Shanghai Yuanxi Instrument Co., Ltd., Shanghai, China).

Five factors (extraction time, liquid-to-material ratio, extraction temperature, alkaline concentration, and extraction repetition) were selected to investigate the extraction process of SEP.

#### 2.2.3. Design of the Response Surface Methodology (RSM) for the Extraction Process

A quadratic polynomial model by Design for Expert 13.0 (Stat-Ease 360, Minneapolis, MN, USA) was developed for the optimization of the SEP extraction based on the results of the single-factor experiments. An analysis of variance was performed to analyze the statistical significance and each term of the fitting model. The 3D surface plots were expressed as the interaction of each variable on the response value (protein content), and the residual distribution plot was expressed as the extent to which the model was affected by disturbance factors.

#### 2.2.4. Artificial Neural Network Modeling

Artificial neural networks (ANN) were used to examine the nonlinear correlation between the input variables and output response. In a MATLAB 2016b (MathWorks Inc., Natick, MA, USA) environment, a total of 87 (29 × 3) data from the RSM were used, of which 70% of the data were used for model training, 15% of the data were used for validation, and the others were used for testing. The mean square error (MSE) was used as the performance determination of the ANN model. The MSE is expressed by the following equation:(2)$$\mathrm{MSE}=\frac{\sum_{i=1}^{r}\left(n_{i}-1\right)S_{i}^{2}}{N-r}$$
where N represents the sample size, r represents the groups, S represents the corrected variance, and N−r represents the degrees of freedom.

#### 2.2.5. Genetic Algorithm to Optimize the ANN Model

The adaptation function of the genetic algorithm (GA) was established by the ANN model of SEP extraction. The process of the GA is shown in Figure 1.

#### 2.2.6. Optimum pH for SEP Precipitation

After extraction at the conditions obtained by the GA, the supernatant was retained by centrifugation at 6793 g for 20 min and divided into 11 equal portions. The pH of the supernatant was adjusted to 1.0, 1.5, 2.0, 2.5, 3.0, 3.5, 4.0, 4.5, 5.0, 5.5, and 6.0, respectively. Then, the supernatant was taken after centrifugation at 10,000 r/min for 15 min, and the protein content was determined using the method described in 2.2.2. The precipitation was dialyzed using a 3.5 kDa dialysis bag for 8 h (pure water was replaced every two hours), and freeze-dried for 24 h to obtain the SEP.

#### 2.2.7. Selenium Content in SEP

The method GB 5009.930-2017 was used to test the selenium content in the SEP with a microwave digestion instrument (MARS6, CEM Corporation, Matthews, NC, USA) and a double-channel atomic fluorescence photometer (AFS-9760, Beijing Haiguang Instrument Co., Ltd., Beijing, China). The result was 125.1499 mg/kg.

#### 2.2.8. Yield of SEP

We referred to the method of Jia et al. [20] with slight modifications to determine the protein content of SEP. An amount of 1 g SER powder was extracted by 50 mL of 5% trichloroacetic acid solution at 25 °C for 1 h to remove the nitrogen of non-protein. The precipitation was centrifuged at 10,615 g for 15 min, and then freeze-dried for 24 h.

The protein content of the SER powder with no nitrogen of non-protein and the purity of the SEP were determined by the Automatic Kjeldahl apparatus (K9860, Zhengzhou Haineng Instrument Co., Ltd., Zhengzhou, China) using the method GB 5009.5-2016, and the protein coefficient was taken as 6.25. The yield of SEP is expressed by the following equation:(3)$$\mathrm{Yield}\;\left(\%\right)=\frac{a\;\times \;b}{c\;\times \;d}$$
where a represents the purity of SEP, b represents the mass of SEP, c represents the mass of selenium-enriched rape powder, d represents the protein content of selenium-enriched rape powder.

### 2.3. Amino Acid Analysis

The amino acid test of the SEP was determined by the amino acid automatic analyzer (L8900, Hitachi High-Tech Co., Ltd., Tokyo, Japan) using the method GB 5009.124-2016.

### 2.4. Antioxidant Activity

#### 2.4.1. DPPH Method

The DPPH ethanol solution was configured by referring to the method of Islam et al. [21] Samples with protein concentrations of 0.25, 0.5, 1, 2, and 4 mg/mL were prepared, and Vitamin C (Vc) was set as a positive control group. The samples were reacted with DPPH ethanol solution for 30 min protected from light, and their absorbance at 517 nm was measured by enzyme marker (1510, Thermo Fisher Scientific Co., Ltd., Waltham, MA, USA). The DPPH radical scavenging rate is expressed by the following equation:(4)$$\mathrm{DPPH}\;\mathrm{Scavenging}\;\mathrm{rate}\;\left(\%\right)=1-\frac{A_{1}-{\;A}_{2}}{A_{0}}\times 100\%$$
where A_1_ represents the absorbance of the control group, A_2_ represents the absorbance of the sample group, and A_0_ represents the absorbance of the blank group

#### 2.4.2. ABTS Method

With reference to the method of Liu et al. [22], the samples with protein concentrations of 100, 200, 400, 800, and 1600 μg/mL were prepared, and the Vc was set as a positive control group. Samples were reacted with ABTS solution for 30 min, protected from light, and their absorbance at 734 nm was measured by enzyme marker (1510, Thermo Fisher Scientific Co., Ltd., Waltham, MA, USA). The ABTS radical scavenging rate was expressed by the following equation:(5)$$\mathrm{ABTS}\;\mathrm{Scavenging}\;\mathrm{rate}\;\left(\%\right)=1-\frac{A_{1}-{\;A}_{2}}{A_{0}}\times 100\%$$
where A_1_ represents the absorbance of the control group, A_2_ represents the absorbance of the sample group, and A_0_ represents the absorbance of the blank group.

### 2.5. In Vitro Cytology Studies

#### 2.5.1. HepG2 Cell Culture and Solution Preparation

The HepG2 cells were cultured in DMEM containing 10% fetal bovine serum (FBS) and 1% penicillin/streptomycin in a constant temperature incubator (Shanghai Xinmiao Medical Equipment Manufacturing Co., Ltd., Shanghai, China) at 37 °C, 5% CO_2_, and 95% saturated humidity. The cells were divided into flasks when cell confluence was 70–80%.

The SEP solution with 1000 μg/L selenium concentration (the protein concentration was 8 mg/mL) was prepared with sterile water before filtering through a 0.22 μm membrane. The samples with selenium concentrations of 25, 50 (daily recommended selenium intake 50–60 μg/d), 100, and 200 μg/L, were prepared in the DMEM medium containing 10% FBS and the SEP solution. The samples were stored in a 4 °C refrigerator.

#### 2.5.2. HepG2 Cytostatic Rate

Cells in the logarithmic growth condition were inoculated in 96-well plates at a density of 5 × 10^3^ cells. Referencing the method of the CCK-8 kit, the HepG2 cells were incubated overnight in a constant temperature incubator at 37 °C, 5% CO_2_, and 95% saturated humidity. The samples with the selenium content of 25, 50, 100, 200, and 400 mg/L were added to the 96-well plate when the culture fluid was aspirated by a pipette. The 5-Fu (100 μg/mL) was set up in the control group, and the DMEM was set up in the blank group. Each group was tested six times after incubation for 24 h. The cells were removed from the incubator and incubated for 1 h with the CCK-8 reagent, and then the absorbance values were measured at 450 nm with an enzyme marker. The cytostatic rate is expressed by the following equation:(6)$$\mathrm{Cytostatic}\;\mathrm{rate}\;\left(\%\right)=1-\frac{{\mathrm{OD}}_{1}-{\;\mathrm{OD}}_{0}}{{\mathrm{OD}}_{2}-{\;\mathrm{OD}}_{0}}\times 100\%$$
where OD_1_ represents the absorbance value of the experimental group (including cells, medium, CCK-8 solution, and SEP solution); OD_2_ represents the absorbance value of the control group experiment (including cells, medium, CCK-8 solution, without SEP solution); OD_0_ represents a blank group (containing medium, CCK-8 solution, without cells and SEP solution)

#### 2.5.3. HepG2 Migration Inhibition Capacity

Cells in the logarithmic growth condition were inoculated in 6-well plates and cultured in an incubator at 37 °C and 5% CO_2_. When the cell confluence was about 90%, three vertical lines were drawn in the culture wells by a 200 μL gun tip. After rinsing twice using PBS solution, samples with selenium concentrations of 25, 50, 100, and 200 μg/L were added. The 5-Fu was set up as the positive control, and the DMEM group was set up as the negative control. The migration rate of the HepG2 cells was observed by an inverted microscope (CKX41, Olympus Co., Ltd., Shinjuku Tokyo, Japan), and the migration of cells at 0, 24, and 48 h was recorded by ImageJ (National Institutes of Health, Bethesda, MD, USA) software. The migration rate is expressed by the following equation:(7)$$\mathrm{Migration}\;\mathrm{rate}\;\left(\%\right)=\frac{H_{1}-{\;H}_{2}}{H_{1}}\times 100\%$$
where H_1_ represents the scratch area at 0 h, H_2_ represents the scratch area at 24 or 48 h.

### 2.6. Statistical Analysis

The data are expressed as the mean ± standard deviation, and all the experiments were performed at least three times. The statistical analysis was performed using one analysis of variance (ANOVA) by Duncan’s multiple comparison test using IBM SPSS statistic 21.0 (International Business Machines Corporation, Armonk, NY, USA) software. The significance level was set at a *p*-value < 0.05. Plotting was performed using GraphPad Prism 8.0 (GraphPad Software, San Diego, CA, USA) and Origin 2022b (OriginLab, Northampton, MA, USA) software.

## 3. Results and Discussion

### 3.1. Selection of the Solvent for Extraction and the Single-Factor Experiments

The effect of extraction solvent and diverse factors such as extraction time, extraction temperature, alkaline concentration, liquid-to-material ratio, and extraction repetition were explored by the single-factor experiments.

Figure 2A shows that the protein content in 0.1 mol/L NaOH solution was significantly higher (*p* < 0.05) than in other systems after 1 h of extraction reacted with the condition of the liquid-to-material ratio of 10:1 and extraction temperature of 50 °C for SER powder. Therefore, the NaOH solution was selected as the system for the extraction of SEP.

For the evaluation of the extraction time, the liquid-to-material ratio was 20:1, the alkaline concentration was 0.1 mol/L, the extraction temperature was 50 °C, and the extraction time was evaluated in a range between 1 and 5 h. Figure 2B shows that the protein content in the solution was significantly increased from 1–3 h (*p* < 0.05), and decreased from 3–5 h with increasing extraction time. This result may have been caused by the reduced protein content in the solution, as the protein was denatured during the long reaction time. Therefore, 3 h was selected as the best length of time for the extraction of SEP.

For the evaluation of the extraction temperature, the liquid-to-material ratio was 20:1, the alkaline concentration was 0.1 mol/L, the extraction temperature was 50 °C, and the extraction temperature was evaluated in a range between 30 and 70 °C. Figure 2C shows that the protein content in the solution increased and then decreased as the extraction temperature increased. The protein content was up to 31.24 mg/g at the extraction temperature of 60 °C. When the extraction temperature was 70 °C, the protein content decreased significantly (*p* < 0.05). This result could be attributed to the fact that the denaturation of the protein was induced by the high temperature, which in turn affected the protein solubilization [20]. Therefore, 60 °C was selected as the optimal temperature for the extraction of SEP.

For the evaluation of the alkaline concentration, the liquid-to-material ratio was 20:1, the extraction time was 3 h, the extraction temperature was 60 °C, and the alkaline concentration was evaluated in a range between 0.1 and 0.3 mol/L. Figure 2D shows that the protein content in the solution rapidly increased when the alkaline concentration was increased from 0.05 mol/L to 0.15 mol/L, and there was a significant difference (*p* < 0.05) between the 0.05 and 0.15 mol/L levels. This result may be caused by the increased protein content in the solution, as the cell structure was destroyed by the alkaline solvent. The rising trend was slow and showed no significant change at 0.15–0.25 mol/L concentration level. With the alkaline concentration at 0.25–0.30 mol/L level, the protein content in the system was significantly decreased (*p* < 0.05). Related studies have shown that amino acid structures such as Lys and Cys, etc., are disrupted at a high alkaline concentration, which in turn leads to a decreased in the selenium content of the SEP [23]. Therefore, 0.25 mol/L was selected as the optimal alkaline concentration for the extraction of SEP.

For the evaluation of the liquid-to-material ratio, the alkaline concentration was 0.25 mol/L, the extraction time was 3 h, the extraction temperature was 60 °C, and the liquid-to-material ratio was evaluated in a range between 10:1 and 70:1 mL/g. Figure 2E shows that there was a little protein in the system when the liquid-to-material ratio was low. The cellulose and other substances in the plant powder absorbed water and swelled, causing the system to be viscous, which resulted in the dissolution of protein molecules slowing down [24]. With the increase in the liquid-to-material ratio, the protein content in the solution was rapidly increased, and when the liquid-to-material ratio reached 60:1 mL/g, the protein content reached a maximum of 53.02 mg/g. Therefore, 60:1 mL/g was selected as the optimal liquid-to-material ratio for the extraction of SEP.

For the evaluation of the extraction repetition, the alkaline concentration was 0.25 mol/L, the extraction time was 3 h, the extraction temperature was 60 °C, the liquid-to-material ratio was 60:1 mL/g, and the extraction repetition was evaluated in a range between 1 and 3 times. Figure 2F shows that the protein content in the solution increased with the increase of extraction repetition, but there was no significant difference between the two and three times. Therefore, two times for the extraction were selected as the optimal extraction repetition for the extraction of the SEP.

### 3.2. RSM Modeling

The purpose of process optimization in this study was to improve the extraction of the SEP. Based on the principle of the Box–Behnken experiment and the results of the single-factor experiments, the extraction temperature (A), extraction time (B), alkaline concentration (C), and liquid-to-material ratio (D) were selected as independent variables to design a 4-factor, 3-level response surface methodology (RSM) model, and the results of RSM model are shown in Table 1. (Since the factor level table of the Design-Expert 13.0 software only allowed setting specific values, the liquid-to-material ratio in the factor experiment table was simplified from X:1 mL/g to X mL/g).

The factors and results in Table 1 were analyzed by multiple quadratic regression fitting using Design-Expert 13.0, and the model regression equation was obtained as Y = 55.58 + 3.56A − 0.9043B − 2.36C + 2.38D + 1.16AB + 0.5888AC − 1.34AD − 1.16BC − 2.07BD − 0.9455CD − 3.37A^2^ − 2.48B^2^ − 5.01C^2^ − 5.16D^2^. The results of the response surface quadratic regression model ANOVA for the SEP extraction process are shown in Table 2.

The effect of the independent variable on the dependent variable was determined by the significance test in the ANOVA [25]. Table 2 shows that the F-value of this model was 27.70, and its *p*-value was less than 0.01, which the model judged to be extremely significant. The *p* value of the out-of-fit was 0.1558 > 0.05, indicating that the model was not affected by the out-of-fit factor. The coefficient of determination R^2^ was 0.9652, and the difference between Adjusted R² and Predicted R² was less than 0.2, indicating that the actual situation was adequately reflected by the model.

The *p* values of factors A (extraction temperature), B (extraction time), C (alkaline concentration), and D (liquid-to-material ratio) in Table 2 were all less than 0.05, indicating that Y (protein content in solution) was significantly influenced by factors A, B, C, and D. The *p* values of factors A, C, and D were less than 0.01, indicating that factors A, C, and D showed extremely significant effects on Y.

The *p* values of the cross terms such as BD, A^2^, B^2^, C^2^, and D^2^ were less than 0.01, indicating that these cross terms showed an extremely significant effect on Y. The rest of the cross terms showed a non-significant effect. Comparing the F-value of the factor, it could be seen that the effect of each factor on the protein content in descending order was A > D > C > B [26]. The normal probabilities distribution of the residuals obtained by Design-Expert 13.0 software is shown in Figure 3.

Figure 3 shows that the probability of residual points in the SEP extraction model was basically distributed on the same straight line. The normal distribution was obeyed by outer residuals, indicating that the actual protein content of the SEP extraction was less different than the predicted content of protein. The 3D surface map and contour analysis of the model are shown in Figure 4.

The effect of the interaction of four factors (A, B, C, and D) on Y was intuitively reflected by observing the steepness, contour density, and contour presentation shape of the 3D surface map [27]. Figure 4 shows that Y tended to increase and then decrease with increasing factor levels when A, B, C, and D interacted with any two factors. The optimal conditions of the RSM model obtained by analysis with the Design-Expert software were that the extraction temperature was 62.5 °C, the extraction time was 3.0 h, the alkaline concentration was 0.25 mol/L, and the liquid-to-material ratio was 62.2:1 mL/g.

### 3.3. ANN Model

ANN is a popular technology and has been applied in all walks of life. The technique of finding relationships between existing data is widely accepted by researchers, including oil extraction [28], preparation of bioactive peptides [29], resveratrol extraction [30], etc.

#### 3.3.1. Hidden Layer Neuron Number Selection

The back propagation artificial neural network (BP-ANN) structure contained the input layer, the hidden layer, and the output layer, where the number of hidden layers and hidden neurons was closely related to the accuracy of the ANN model. The results of the RSM model showed that the protein content in the system was significantly affected (*p* < 0.05) by extraction temperature, extraction time, alkaline concentration, and liquid-to-material ratio. Therefore, the input layer neurons of the BP-ANN were 4, and the output layer neuron was 1. A single hidden layer was used to model the network topology, and the network topology was 4-X-1.

Using the tool Neural Network Fitting in MATLAB 2016b, 87 groups of data from the response surface model were randomly selected to build the BP-ANN model, where the data of 70% (61 groups) were trained in the model, and the data of 30% (15% + 15%, 13 + 13 groups) were validated and tested the model. The numbers of hidden layer neurons were selected in the range of 3–12, and the model performance was determined by the neural network MSE. The corresponding relationships between the number of hidden layer neurons, model MSE values, and the number of iterations is shown in Table 3.

Table 3 and Appendix A show that when the number of neurons in the hidden layer was 7, the MSE of validation reached a minimum value of 4.670 × 10^−3^. The function of the model converged at 20 epochs with the MSE tending to be stable. These results indicated that the model possessed excellent generalization ability when the neural network topology was 4 × 7 × 1. Therefore, 7 was selected as the number of neurons in the hidden layer of the BP-ANN for the SEP extraction.

#### 3.3.2. BP-ANN Model Training Correlation Evaluation

A portion of the data were used to train the BP-ANN model and the remaining portion was used for testing and validating the model [27]. The relationship of the regression fit between the actual and predicted values are shown in Figure 5. The results, solid and dashed lines, indicate that an excellent linear fit between the real and predicted values was observed, (In Figure 5, the solid line basically coincides with the dashed), and the correlation coefficient of this model was 99.98%.

#### 3.3.3. Genetic Algorithm to Optimize BP-ANN Model

The genetic Algorithm (GA) was an evolutionary algorithm that drew on the ideas of Darwinian natural evolution and a genetic mechanism to search the globe and found the optimal solution by simulating the natural evolutionary process [31]. In this study, after setting the basic GA parameters (population size NP = 50, crossover probability Pc = 0.8, variation probability Pm = 0.2, and individual length L = 4), the ANN model was invoked as a fitness function and “protein content” was used as the fitness value with a maximum epoch of 100.

The optimal parameters were found by the GA through continuous iteration, and the population’s optimal fitness value during an iteration is shown in Figure 6. In the beginning condition, the population search property of GA possessed a clear effect, and the selected individuals underwent a sharply upward change and were chosen. After running the program, a series of operations including cross, selection, variation, etc., were performed by the GA, and the fitness function converged when 78 iterations were performed. When the iterations of the GA proceeded to 100 epochs, the GA stopped running. The results of the GA were that the predicted content of protein was 58.04 mg/g, the extraction temperature was 59.4 °C, the extraction time was 3.0 h, the alkaline concentration was 0.24 mol/L, and the liquid-to-material ratio was 65.2:1 mL/g.

### 3.4. Comparative Analysis of RSM and ANN Models

#### 3.4.1. Comparison of Model Error Analysis

Table 4 shows the error analysis between the RSM and ANN. The predictive capability of the ANN model was reflected by calculating and comparing the correlation coefficient R^2^, root means square error (RMSE), mean absolute deviation (MAD) and spherical probability error (SPE) [17]. The calculation is shown as the following equation:(8)$$R^{2}=\frac{{\left(\sum_{i=1}^{n}\left(Y_{1}\;-\;{\bar{Y}}_{1}\right)\left(Y_{1}\;-\;{\bar{Y}}_{p}\right)\right)}^{2}}{{\left(\sum_{i=1}^{n}\left(Y_{1}\;-\;{\bar{Y}}_{1}\right)\left(Y_{p}\;-\;{\bar{Y}}_{p}\right)\right)}^{2}}$$
(9)$$\mathrm{RMSE}={\left(\frac{1}{n}\sum_{i=1}^{n}{\left(Y_{p}-Y_{1}\right)}^{2}\right)}^{\frac{1}{2}}$$
(10)$$\mathrm{MAD}(\%)\;=\;\frac{1}{n}\sum_{i=1}^{n}\frac{\left|Y_{1}\;-\;{\bar{Y}}_{1}\right|}{\left|Y_{1}\right|}\times 100\%$$
(11)$$\mathrm{SPE}\;(\%)\;=\;\frac{\mathrm{RSME}}{{\bar{Y}}_{1}}\times 100\%$$
where n represents the number of samples, Y_1_ represents the test value, ${\bar{Y}}_{1}$ represents the test sample mean, Y_p_ represents the model predicted value, and ${\bar{Y}}_{p}$ represents the model predicted mean.

The RMSE, MAD, and SPE were low, and the model correlation coefficient R^2^ was high, indicating that the model possessed high accuracy and reliability [32]. Table 4 shows that the R^2^ of the ANN model was higher than the RSM, and the RMSE and SPE values are lower than the RSM indicating that the ANN model was better than RSM in prediction accuracy. Interestingly, the MAD value of the ANN model was slightly higher than the RSM, indicating that the ANN model possessed a greater degree of dispersion compared to the RSM model. Figure 7 shows the deviation of the predicted data from both models compared to the real data, to further explore the reliability of the models.

#### 3.4.2. Prediction Data Comparison

Figure 7 shows that the ANN predicted data were usually close to the experimental values indicating that the prediction accuracy of the ANN model was higher than the RSM model. Interestingly, the results of the 14th and 29th group were different from the others. This is the reason why the ANN model MAD in Section 3.4.1 was higher than the RSM.

#### 3.4.3. Comparison and Validation of the Optimal Solutions of the Models

The protein was extracted by the predicted conditions of the two models, and the results are shown in Table 5. The protein content of 57.69 mg/g predicted by the GA + ANN was higher than the 55.31 mg/g predicted by the RSM model, and the relative error was lower than the RSM model. Therefore, it can be concluded that the GA + ANN model was a more effective method than the RSM for optimizing the process of SEP extraction. This result is consistent with the findings of Hee-Jeong [32].

### 3.5. Optimal pH for SEP Precipitation

The optical density (OD) value of the supernatant after centrifugation was determined by the Coomassie brilliant blue method, which positively correlated with the extracted protein content in the solution [20]. Therefore, the pH with the lowest protein content in the supernate of the SEP was the optimum pH for precipitating the SEP. Figure 8 shows that the OD value in pH = 3.5 of the supernate was the lowest, indicating that the sedimentation of the SEP was relatively complete. Hence, the optimum pH for SEP precipitation was 3.5.

### 3.6. The Yield of SEP

The purity of the SEP was determined by the Kjeldahl method, and the protein content of the SEP was the purity of the SEP. Figure 9 shows that the protein content of the SER powder was 13.31%, and the purity of the SEP was 54.23%. The yield of the SEP was calculated to be 61.71% with the optimal conditions of the GA after twice extraction.

### 3.7. Amino Acid Analysis

The amino acid analysis results of the SEP are shown in Table 6 (Trp was destroyed). The 16 amino acids in the SEP were measured, including seven essential amino acids and accounting for 35.23% of the total amino acids, which was higher than the standard of high-quality protein (12.7%) proposed in the reports of the WHO and FAO [33]. Among the amino acid composition of SEP, the content of Glu (12.378%) and Asp (5.302%) were the two highest amino acids in the amino acid composition of SEP, which are critical sources of energy for the small intestinal epithelial cells of mammals, and possess bio-activities such as maintaining the intestinal function, relieving oxidative stress, and neuromodulation [34]. Studies have shown that the activity of antioxidant peptides produced by protein hydrolysis are influenced by hydrophobic amino acid residues [35], and the hydrophobic amino acid percentage of the SEP was 36.85%. It could be deduced that the SEP possessed high nutritional value and holds potential for a functional product.

### 3.8. Antioxidant Activity

The antioxidant activity of the SEP is shown in Figure 10, and the antioxidant activity of the SEP increased with increasing protein concentration. DPPH radicals were significantly scavenged by the SEP at concentrations ranging from 0.25 to 2 mg/mL (*p* < 0.05). The DPPH radical scavenging rate reached 86.8% when the SEP concentration was 4 mg/mL. The scavenging rate of ABTS radicals was significant (*p* < 0.05) for all the protein concentration levels. When the SEP concentration reached 1600 μg/mL, the scavenging rate of the ABTS radicals reached 99.65% with no significant difference in V_C_.

The antioxidant activity of the SEPs was not only related to the hydrophobic amino acids in the protein but also related to the selenium content of the protein. Xiang et al. [36] compared the selenium content and antioxidant activity of two selenium-containing proteins after purification and concluded that the higher the selenium content in the protein the better its free radical scavenging ability. Zhu et al. [37] showed that selenium-enriched peptides had better antioxidant activity than non-selenium-containing peptides, and the selenium content in the peptides was positively correlated with the antioxidant capacity. In this study, it can be concluded that the antioxidant capacity of the SEP increased with the increase in the SEP concentration.

Another bioactivity of selenium is anticancer activity. In this study, the anticancer activity of SEP was preliminarily explored by using an in vitro cytological method.

### 3.9. In Vitro Cytological Study of Selenium-Enriched Rape Protein

#### 3.9.1. HepG2 Cytostatic Rate

Excessive consumption of Se can lead to poisoning. The recommended daily intake of selenium was 50–60 μg/d, and the maximum intake was 400 μg/d according to the Chinese Dietary Guidelines. In this study, the selenium content of the SEP was 125.1499 mg/kg ≈ 125 μg/g. Therefore, HepG2 was treated in the SEP solutions with selenium concentrations of 25, 50, 100, 200, and 400 μg/L (protein concentrations of 0.2, 0.4, 0.8, 1.6, and 3.2 mg/mL, respectively) for 24 h, and the 5-Fu (100 μg/mL) was set as a positive control group. The changes in cytostatic rates are shown in Figure 11.

The cytostatic rate of HepG2 was significantly increased (*p* < 0.05) with an increase in selenium concentration. Figure 11 shows that the cytostatic rate of the 400 μg/L level was only 4.48% lower than the 5-Fu level, indicating that the SEP possessed a good inhibitory effect on HepG2 cell proliferation.

#### 3.9.2. Effect of the SEP on the Migration of HepG2 Cells

The effect of the SEP on the HepG2 migration ability was investigated by using the scratch test, and the results are shown in Figure 12. The migration capacity of the HepG2 cells at 24 h and 48 h decreased significantly (*p* < 0.05) with the increasing selenium concentration of the sample. Among them, the migration rate of the HepG2 cells decreased by 50.62% at 24 h and by 28.74% at 48 h compared with the negative control group (DMEM). Interestingly, when the selenium concentration of the SEP was 25 μg/L, the migration rate at both 24 and 48 h was slightly higher than the DMEM group. The reason for this result may have been the low selenium concentration of the 25 μg/L group; the protein in the system provided the nutrients to the HepG2 cells.

## 4. Conclusions

In this study, a 4 × 7 × 1 topology structure of ANN was used to develop a model for the optimization of the SEP’s extraction, and the GA was used to globally find the function that would obtain the optimal process parameters. Upon comparing the model error, the difference between the predicted data and the real data, and the verification test of the optimal solution, it was revealed that the GA + ANN model was better than RSM for the predictive accuracy of the SEP extraction. Meanwhile, the SEP was obtained by applying the optimized extraction process, and the antioxidant and anticancer activities of the SEP were investigated systematically, and we found their inhibitory effect on HepG2 cells’ proliferation and migration was enhanced as selenium concentration increased. This research provided a new perspective on SEP’s extraction and bioactivity exploration, and we found that the ANN was a practical program when it was used to build models. However, the ANN cannot design a random experimental program for researchers because the function of experimental design was not developed. Therefore, a combined method, including RSM, ANN, and optimization algorithms, could be used to fully utilize the advantages of each technology in the future.

## Figures and Tables

**Figure 1 foods-11-03823-f001:**
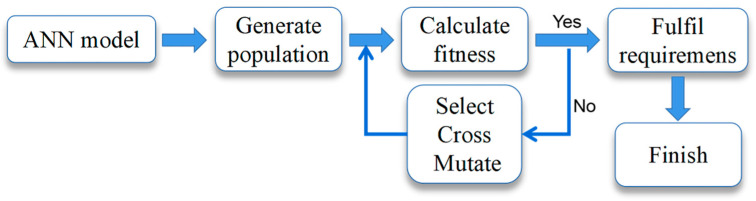
Genetic algorithm process.

**Figure 2 foods-11-03823-f002:**
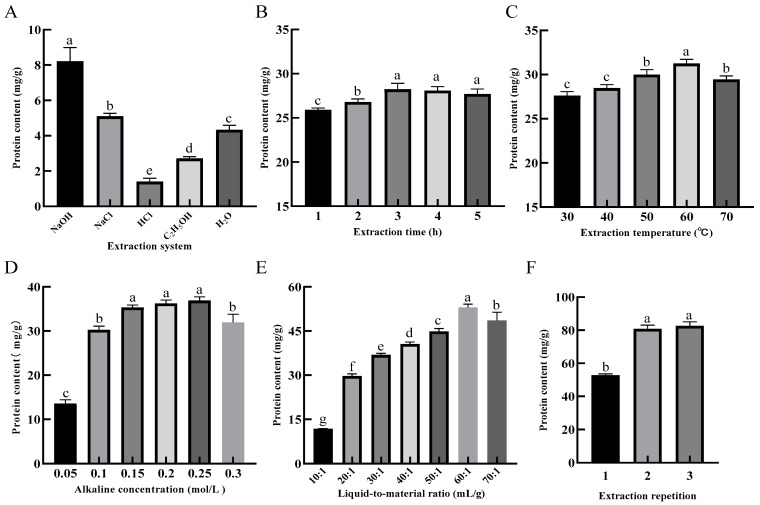
Effects of extraction system (**A**), extraction time (**B**), extraction temperature (**C**), alkaline concentration (**D**), liquid-to-material ratio (**E**) and extraction repetition (**F**) on the extraction rate of protein. Different lower-case letters in the column charts indicate a significant difference at *p* < 0.05.

**Figure 3 foods-11-03823-f003:**
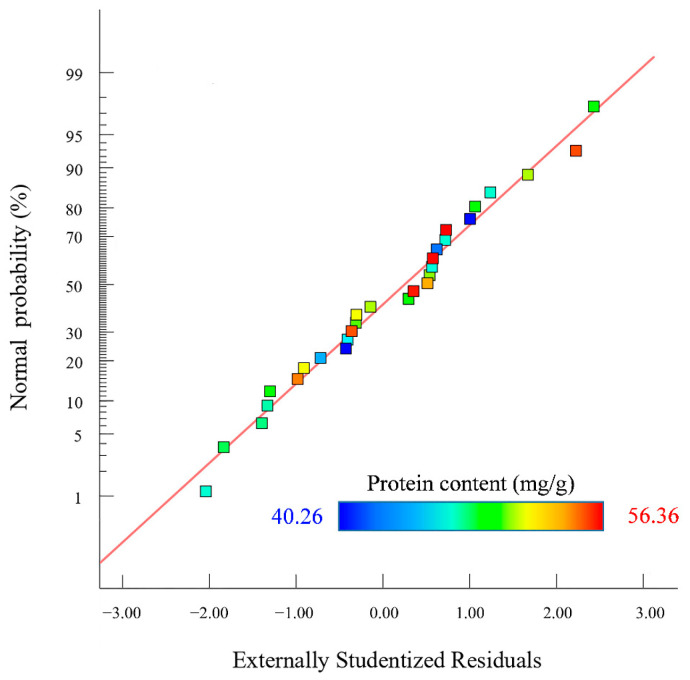
Normal probability distribution of residuals.

**Figure 4 foods-11-03823-f004:**
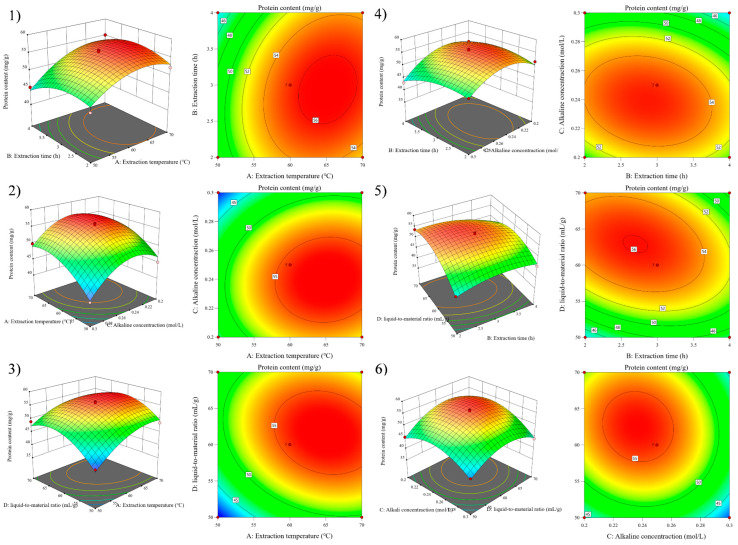
Response surface graphs (**1**–**6**) show the effect of the interaction of various experimental factors on protein content.

**Figure 5 foods-11-03823-f005:**
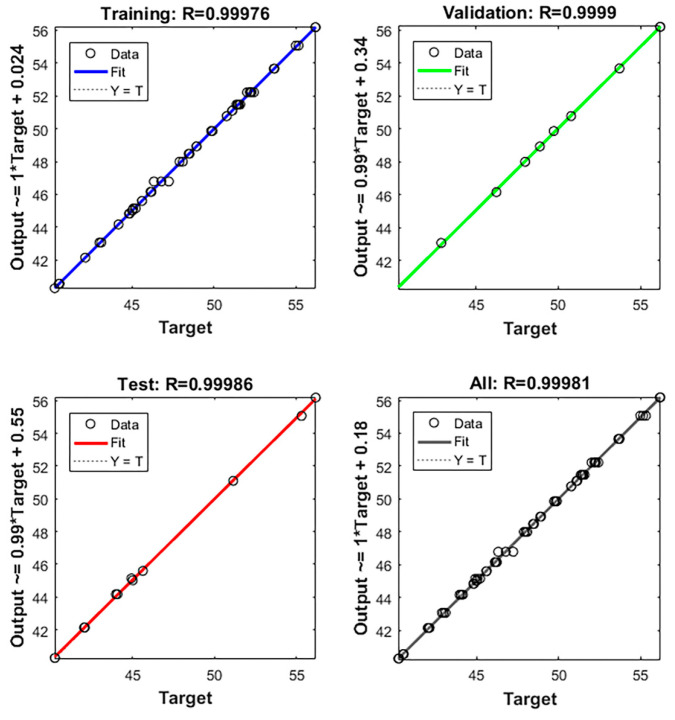
Fitted curves of the SEP from BP-ANN model training.

**Figure 6 foods-11-03823-f006:**
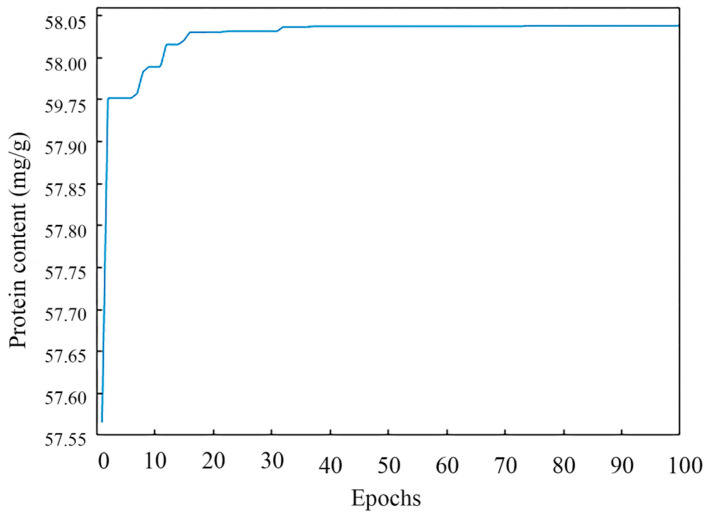
The genetic algorithm convergence process.

**Figure 7 foods-11-03823-f007:**
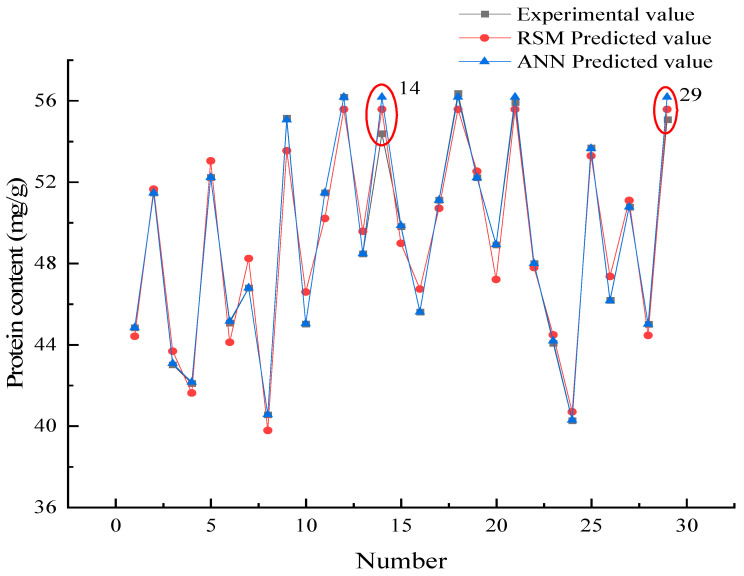
Comparison of the predicted data and real data.

**Figure 8 foods-11-03823-f008:**
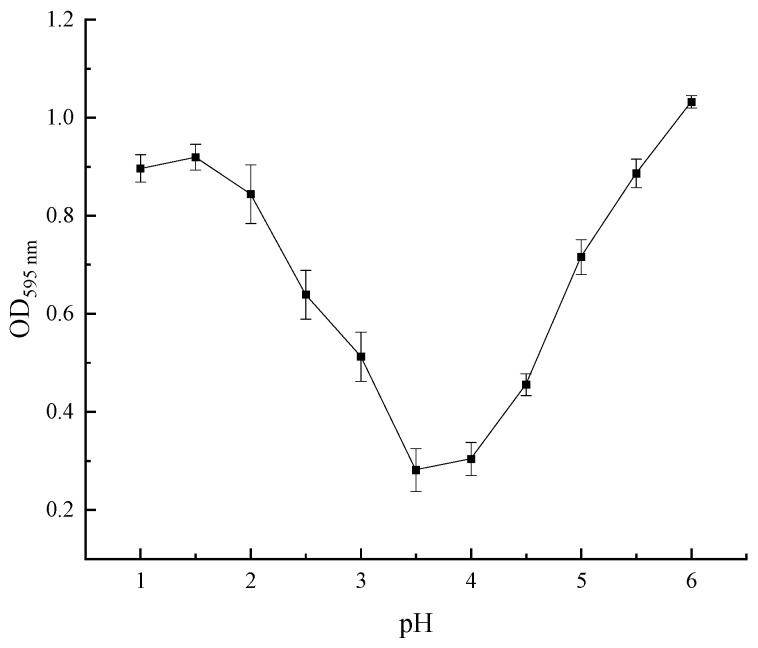
Effect of pH on the optical density of the supernate.

**Figure 9 foods-11-03823-f009:**
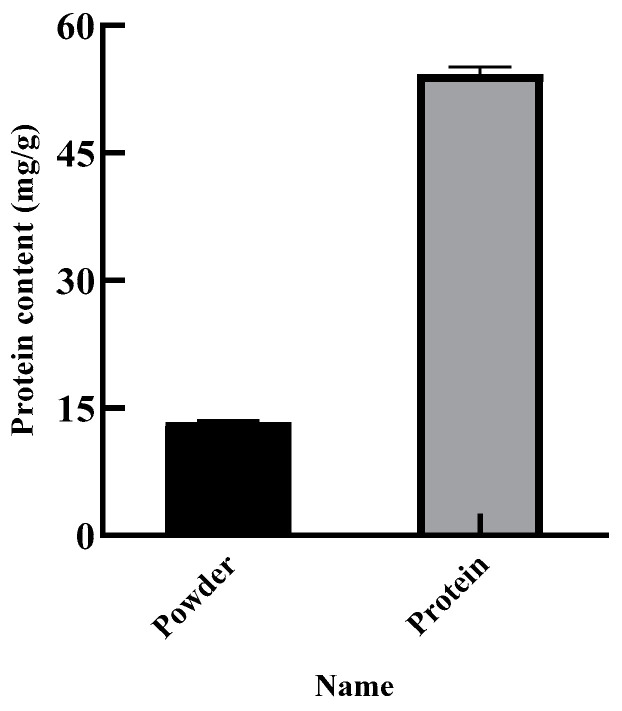
Content of SER powder and SEP.

**Figure 10 foods-11-03823-f010:**
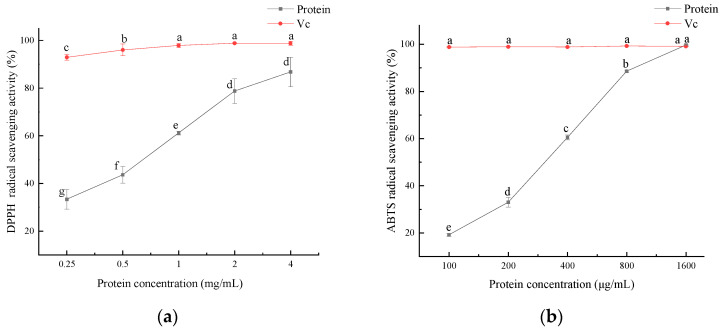
Antioxidant activity of SEP protein. (**a**) Effect of SEP on DPPH radical scavenging; (**b**) Effect of SEP on ABTS radical scavenging. Notes: means with different lower-case letter differed significantly at *p* < 0.05.

**Figure 11 foods-11-03823-f011:**
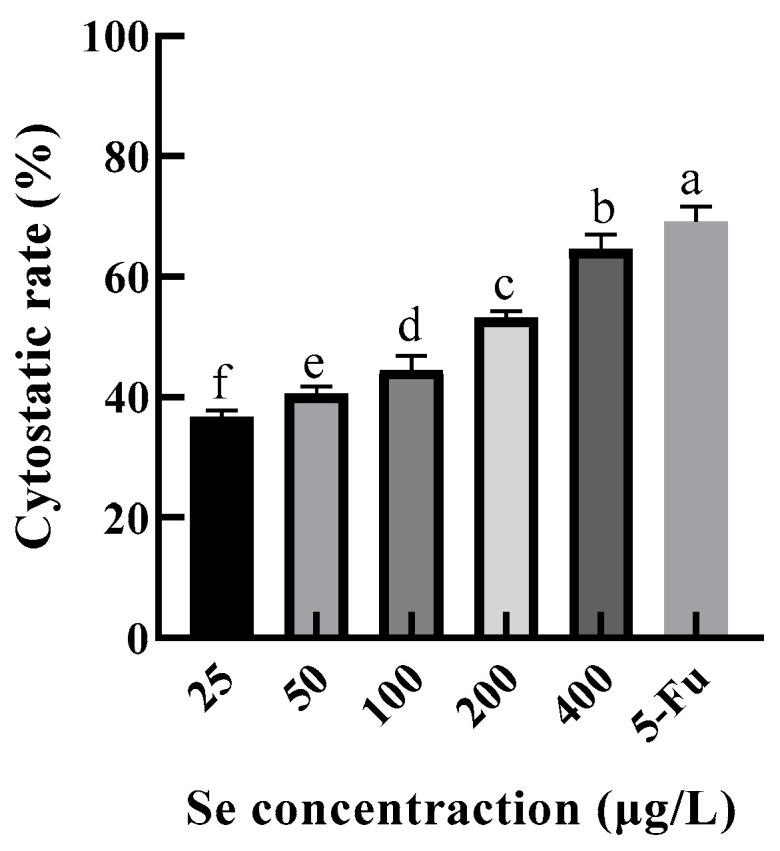
Inhibitory effect of selenium-enriched rape protein on HepG2 cells. Notes: means with different lower-case letter differed significantly at *p* < 0.05.

**Figure 12 foods-11-03823-f012:**
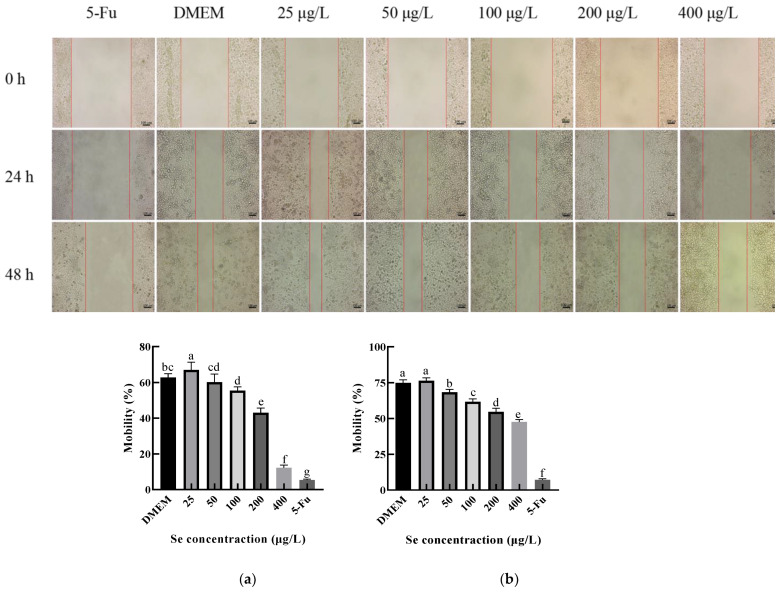
The effect of the SEP on the migration ability of the Hep G2 cells detected by scratch test. (**a**) Effect of the SEP on 24 h migration inhibition of the HepG2 cells; (**b**) Effect of the SEP on 48 h migration of the HepG2 cells. Notes: means with different lower-case letter differed significantly at *p* < 0.05.

**Table 1 foods-11-03823-t001:** Design scheme and result of response surface test.

No.	Factor				Protein Content mg/g	
	A °C	B h	C mol/L	D mL/g	Actual Value	Predicted Value
1	60	2	0.25	50	44.83	44.41
2	70	3	0.25	70	51.46	51.65
3	60	4	0.3	60	43.01	43.67
4	60	3	0.3	50	42.09	41.62
5	70	2	0.25	60	52.23	53.04
6	50	4	0.25	60	45.07	44.11
7	50	2	0.25	60	46.79	48.24
8	50	3	0.25	50	40.55	39.77
9	70	4	0.25	60	55.15	53.55
10	50	3	0.2	60	45.02	46.59
11	60	2	0.2	60	51.47	50.21
12	60	3	0.25	60	56.18	55.58
13	70	3	0.25	50	48.47	49.58
14	60	3	0.25	60	54.38	55.58
15	70	3	0.3	60	49.81	48.99
16	60	4	0.25	50	45.61	46.73
17	60	4	0.2	60	51.11	50.71
18	60	3	0.25	60	56.36	55.58
19	70	3	0.2	60	52.22	52.54
20	50	3	0.25	70	48.91	47.21
21	60	3	0.25	60	55.92	55.58
22	60	2	0.3	60	47.99	47.79
23	60	3	0.3	70	44.07	44.48
24	50	3	0.3	60	40.26	40.69
25	60	2	0.25	70	53.68	53.30
26	60	4	0.25	70	46.18	47.35
27	60	3	0.2	70	50.77	51.10
28	60	3	0.2	50	45.01	44.45
29	60	3	0.25	60	55.08	55.58

**Table 2 foods-11-03823-t002:** Variance analysis of regression model.

Source	Sum of Square	df	Mean Square	F	*p*	Significance
Model	643.04	14	45.93	27.70	<0.0001	**
A	152.24	1	152.24	91.82	<0.0001	**
B	9.81	1	9.81	5.92	0.029	*
C	67.05	1	67.05	40.44	<0.0001	**
D	67.76	1	67.76	40.87	<0.0001	**
AB	5.37	1	5.37	3.24	0.0936	
AC	1.39	1	1.39	0.8365	0.3759	
AD	7.20	1	7.2	4.34	0.0559	
BC	5.34	1	5.34	3.22	0.0943	
BD	17.10	1	17.1	10.31	0.0063	**
CD	3.58	1	3.58	2.16	0.1641	
A^2^	73.78	1	73.78	44.50	<0.0001	**
B^2^	39.77	1	39.77	23.99	0.0002	**
C^2^	162.90	1	162.9	98.25	<0.0001	**
D^2^	172.76	1	172.76	104.20	<0.0001	**
Residual	23.21	14	1.66			
Lack of fit	20.43	10	2.04	2.93	0.1558	
Pure error	2.79	4	0.6967			
Cor total	666.26	28				
R^2^ = 0.9652	Adjusted R^2^ = 0.9303	Predicted R^2^ = 0.8169	Std. Dev. = 1.29			

* Represents values have a significant difference, *p* < 0.05; ** represents values have an extremely significant difference, *p* < 0.01.

**Table 3 foods-11-03823-t003:** Training result for BP artificial neural network with different neurons.

Network Topology	MSE	Epoch
4 × 4 × 1	9.658 × 10^−2^	16
4 × 5 × 1	5.647 × 10^−3^	25
4 × 6 × 1	5.403 × 10^−3^	18
4 × 7 × 1	4.670 × 10^−3^	20
4 × 8 × 1	7.864 × 10^−3^	10
4 × 9 × 1	5.253 × 10^−3^	7
4 × 10 × 1	6.512 × 10^−3^	8
4 × 11 × 1	1.0515 × 10^−2^	10

**Table 4 foods-11-03823-t004:** Comparison of parameters between RSM and ANN.

Model	R^2^	RMSE	MAD (%)	SPE (%)
RSM	0.9652	0.8947	4.0177	1.83
ANN	0.9998	0.3986	4.2636	0.81

**Table 5 foods-11-03823-t005:** Comparison and verification of different models to extract SEP.

Model	Temperature (°C)	Time (h)	Alkali Concentration (mol/L)	V/M (mL/g)	Predicted (mg/g)	Actual (mg/g)	Std. Dev. (%)
RSM	62.5	3.0	0.25	62.2:1	56.50	55.31	2.15
ANN	59.4	3.0	0.24	65.2:1	58.04	57.69	0.61

**Table 6 foods-11-03823-t006:** Analysis results of SEP amino acids.

Types of Amino Acids	Amino Acid Content (%)	Types of Amino Acids	Amino Acid Content (%)
Asp	5.302 ± 0.054	Ile *	2.406 ± 0.025
Thr *	2.384 ± 0.068	Leu *	3.644 ± 0.060
Ser	2.262 ± 0.110	Tyr	1.806 ± 0.020
Glu	12.378 ± 0.042	Phe *	3.166 ± 0.008
Gly	2.602 ± 0.014	Lys *	2.900 ± 0.028
Ala	2.948 ± 0.011	His	1.498 ± 0.003
Cys	/	Arg	2.590 ± 0.088
Val *	3.854 ± 0.201	Pro	3.202 ± 0.025
Met *	0.460 ± 0.006		
total	53.402 g/100g		

* Represents essential amino acid. “/” means that Cys was not detected as it was destroyed during the process of amino acid hydrolysis.

## Data Availability

Data are not available in public datasets; please contact the authors.

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
