# Peer review of "Comparison of an Artificial Neural Network and a Response Surface Model during the Extraction of Selenium-Containing Protein from Selenium-Enriched Brassica napus L."

_foods, 2022, doi:10.3390/foods11233823_

Round 1

Reviewer 1 Report

The authors have comprehensively studied the optimization of extraction of selenium-enriched rape protein by using artificial neural networks and response surface models. The work done encompasses both experimental and computational research, with the experimental work used for validation and model design. Besides testing the efficiency of the optimization protocol the authors evaluated the antioxidant and anti-cancer properties of the selenium-enriched protein. This was done using amino-acid analysis, antioxidant testing, and cytological studies with HepG2 cells. The authors have successfully derived the optimal parameters of temperature, alkaline concentration, and liquid-to-material ratio, with the data possessing statistical significance. Besides this, they justified the use of selenium-enriched protein using several methods.

Despite this, I would recommend a major revision for several reasons.

- The English language and formation of sentences are not adequate and take away from the work done as the message of the article is often unclear. This is seen particularly with the first part of the article (abstract, introduction)

- Limitations of the computational methods are not discussed

- There is no discussion about the applicability of the developed model to other similar extraction processes

L13: Was obtained using the genetic algorithm is more appropriate.

L14: The abbreviations DPPH, ABTS and CCK-8 are introduced before fully explaining these words.

L23: I think that instead of centration you meant concentration.

L23: “a powerful antioxidant ability” is not the correct form; please fix this by rewriting the sentence.

L30: Rephrase the part after “due to its bioactive functions” as antioxidants; anticancer is not the correct form.

L37: Massive is not the appropriate word; major is better. If you mean with iron death an iron-induced cell death, then please rewrite for greater clarity.

L38: Selenium does not have to be spelled with the upper-case

L38: The sentence about consumer awareness is poorly written; please correct it.

L48: Explain in more detail was the SCP (I'm assuming protein) involved in the formation in a negative or positive way; this makes a big difference.

L48: What is SCP? You did not disclose the full meaning behind this abbreviation

L63: I think you mean the primary method of optimization; please rewrite this sentence as it is not clear.

L67: Disclose what limitations you are referring to so the reader knows why you chose specific methods

L91: Use lowercase for the first letter of selenium.

L168: Use lowercase for the first letter of nitrogen.

L173: The equation has a typo, yield instead of yield

L196: Start the sentence with a large capital (The)

L297: Use “and swell” instead of “to swell”

L311: Use “The purpose of process optimization” instead of “The purpose of the optimum process”

L389: The A in “A part of the data” is redundant; just write “Part of the data…”

L392: Solid and dashed lines? Please specify

L407: This sentence is not clear; please rewrite it to make it more clear in terms of what you are trying to communicate to the reader

L419: Why is “was reflected by” in red? Was this a markup not corrected?

L442: The sentence starting with “Figure 7… “does not make sense and should be split into two sentences.

L464: This sentence is not correct; after SEP why is there is. Rephrase it completely.

L465: Power or powder? Correct accordingly.

L475: This sentence is not correct; please correct both the part with amino-acid content and do not use past tense with something that is generally true (epithelial cells of mammalian…)

L495: This sentence does not make sense. Please correct it.

L526: Rewrite the last part of the sentence to “…with increasing selenium concentration of the sample” (skip the as it is redundant)

L543: However is used twice. Please explain in more detail how the ANN model could not help the researchers.

L553: Start the sentence with a capital letter (We)

L555: Sample concentration of what? Be specific

L556: “Of SEP solution”, you don’t need to use “of the SEP solution”. The same applies to L557.

L560: Add “of” before selenium concentration is used in the sentence.

Reviewer 2 Report

The manuscript compared the models made from an artificial neural network and response surface methodology (RSM) in extracting selenium-containing protein from Brassica napus. It is an interesting work but several points are unclear. The specific comments are as follows:

Abstract: the abbreviation of the ABTS, CCK-8, DPPH, and so on should be explained with their full name, especially, when it first appears in the manuscript.

Line 48: What is the meaning of SCP? In Line 52 and Line 64, it states using SCP extraction, and in Line 58, the meaning of the sentence seems that SCP extraction had a better efficiency and yield compared to the enzymatic and ultrasound-assisted methods. The authors stated that the protein could easily be denatured when the ultrasound-assisted applied. Although the lack of introduction of SCP extraction, the temperature during extraction employed was up to 70 oC, I am wondering how the authors avoid the protein denatured when the extraction temperature was so high. 

The introduction must be improved by extensively introducing what SCP extraction is and its application in food relative areas.

Line 67:  According to the meaning of the following sentence, I guess you would like to say "However" rather than "Simultaneously".

Line 72: ANN?

Line 87: please change ";" to ":"

Line 111: Please use the equation mode and mark all the equations with numbers in the manuscript.

Line 123: The range (i.e. minimum and maximum) of the input five factors for RSM should be displayed as a table here. 

As they all show antioxidant activity, the differences between DPPH and ABTS should be clarified.

Line 250: the discussion is missing in this section. Many places only displayed the results but merely have deep discussions (e.g. the reason to cause the phenomenon). It should be improved by deep discussion.

Line 333: Add the space after R2

Line 343: the authors judged the weight importance based on the value of F values, do you have any reference to support this? It is usually to use the corresponding coefficient of the factor 

Line 345: Please delete "." after "order"

Figure 10 (a) I am wondering why there are significant differences of Vc between the protein concentration of 1 and 0.5/0.25.

The resolutions of Figures 3 and Line 533 are not good, please improve them. Why aren't there any captions of Figure in Line 533? Its content seems different from Figure 12.

Reviewer 3 Report

Reviewed manuscript has got innovative data on protein extraction from rape biomass. Protein extract from plants enriched in selenium could be valuable food components, so subject of the study fits the journal scope. I propose:

1)      In abstract – CCK-8 shows proliferation inhibition,

2)      Correct both sentences in lines 40-44 (in first one should be organic, in second “..egg laid by chickens.. – by hens will be more correct”,

3)      In whole text – please use small letter in word  selenium,

4)      I suggest change factor “extraction times” into extraction repetition or extraction cycle to differentiate from extraction time,

5)      Centrifugation speed should be presented as multiplication of g, not rotation per min,

6)      Please correct unit in line 217, it should be µg/L,

7)      Figure 9 presents values 1.331% and 5.423% - not as mentioned in lines 465-466,

8)      Please give full names of Vc in figure 10 and 5-Fu in figure 11.

Round 2

Reviewer 1 Report

The authors successfully resolved all issues raised by this reviewer. Consequently, the manuscript has been significantly improved.

Author Response

Thank you for your efforts in reviewing our mansuscript

Reviewer 2 Report

The authors addressed most of my comments, but there are some parts that should be modified before it can be potentially accepted. For example, in the reply of the reviewer, I mentioned whether the authors mean "ANN?" in Line 77 in the revised version. The authors spelled "AAN model"

According to the contents, I guess section 3. should be Results and Discussion, marked in Line 262, and in Line 549 should be Conclusion. I am not sure if the authors have clearly distinguished the discussion and conclusion or not, the discussion is the explanation of the result observation, and the conclusion should be contained the main finding based on the results and use be concise. Obversely, Lines 558-561 should be moved to the discussion above. Please shorten the conclusion and show off the main finding. Please be careful next time.
